# The Connections among Interacting with Nature, Nature Relatedness and Dietary Choices: A Pilot Mixed Methods Study

**DOI:** 10.3390/ijerph21070899

**Published:** 2024-07-10

**Authors:** Dahlia Stott, Chetan Sharma, Jonathan M Deutsch, Brandy-Joe Milliron

**Affiliations:** 1Department of Health Sciences, College of Nursing and Health Professions, Drexel University, Philadelphia, PA 19104, USA; dps85@drexel.edu (D.S.); csharma@tnstate.edu (C.S.); st96d633@drexel.edu (J.M.D.); 2Department of Food and Animal Sciences, Tennessee State University, Nashville, TN 37209, USA

**Keywords:** nature, interactions with nature, nature relatedness, connection to nature, dietary intake, dietary choices, sustainable food choices

## Abstract

Interacting with nature and a connection to nature (Nature Relatedness [NR]) have been associated with better mental and physical health as well as higher levels of physical activity. However, limited research has explored how interacting with nature and NR may promote healthful dietary behaviors. The purpose of this pilot convergent mixed methods study was to explore the connections between interacting with nature, NR, and dietary choices. For descriptive purposes, we measured participants’ (*n* = 25) interactions with nature, NR (total, experience, perspective, and self) scores, physical activity, and Healthy Eating Index-2020 scores. In-depth interviews (*n* = 13) explored the factors that influence interacting with nature and food choices. Quantitative and qualitative data were analyzed separately, and later integrated to yield a more complete understanding of the research inquiry than either form of data would have provided alone. The results from this pilot mixed methods study indicate that there are connections between the variables of interest and that NR may play a pivotal role in how interacting with nature may promote sustainable dietary behaviors. These findings could promote the potential for utilizing nature-based experiences to improve dietary intake.

## 1. Introduction

Emerging research suggests that interacting with nature is associated with better mental and physical wellbeing, cognitive health, and greater engagement with physical activity [1,2,3,4,5,6,7]. Three types of interactions with nature have been proposed by Keniger et al. (2013): intentional, incidental, and indirect [8]. Intentional interactions with nature have been defined as purposeful interactions with nature such as going to a park. Incidental interactions with nature can be defined as experiencing nature secondary to the main activity being performed and includes being in the presence of indoor plants or walking through a garden while on an errand. Indirect interactions with nature can be defined as experiencing nature without being in nature, such as looking at nature through a window. Each of these types of interactions with nature has been associated with positive mental and physical health statuses and engagement with healthy behaviors [5]. For example, Shanahan et al. (2016) suggest that if all individuals spent 30 min in nature (or intentionally interacting with nature), the cases of depression and high blood pressure would decrease by nearly 10% [4]. The authors of intervention studies have reported that after a nature-based interventions, mental and physical health improvements were observed [9,10,11,12]. The benefits of spending time in nature are not limited to spending time in an outdoor natural area. Incidentally interacting with nature by having houseplants in the home has been associated with mental wellbeing [13,14]. Having a view of nature at home, or indirectly interacting with nature, has been associated with greater life satisfaction and self-reported health and decreased depression and anxiety [2,15,16].

Connection to nature can be defined as an individual’s relationship with nature, including their feelings about, values of, and behaviors in nature [17]. While connection to nature was originally used to measure environmental stewardship, it has been adopted and utilized by health researchers [18]. Nature Relatedness (NR), a measurement of connection to nature, was first introduced by Nisbet and colleagues and consists of three subscales: he experience, perspective, and self [18]. NR Experience measures an individual’s physical familiarity with nature. NR Perspective is used to assess an individual’s personal external world view of nature. NR Self measures an individual’s personal connection to nature. Numerous researchers have reported that NR is also associated with positive mental health, physical health, and engagement with physical activity [19,20,21,22,23].

Despite the known health benefits of interacting with nature and connecting to nature, limited research has explored how interacting with and connecting to nature may influence dietary behaviors. Milliron et al. (2021) were the first to report greater fruit and vegetable consumption and dietary diversity among the individuals who report a higher NR score [24]. Understanding the connections among interacting with nature, NR, and dietary choices is important because dietary choices influence personal health. Consuming a diet rich in fruits and vegetables and limited in ultra-processed foods is associated with a lower risk for the leading causes of death: cardiovascular disease, type two diabetes mellitus, and cancer [25,26,27,28]. The purpose of this study was to explore the interconnections among interacting with nature, NR, and dietary choices using a mixed methods approach. The findings from this research may be used to inform the development of nature-based interventions to promote time spent in nature, NR, and a healthy diet.

## 2. Materials and Methods

### 2.1. Study Design

Using a convergent mixed methods design, this pilot study collected descriptive quantitative data from a survey and dietary recalls and qualitative data from in-depth interviews (Figure 1). Both sets of data were collected and analyzed separately, and then results were integrated together to have a better understanding of the research question. This methodology was used to allow for the strengths of quantitative (objective measurements) and qualitative methods (details, personal experiences, etc.) to complement each other and provide more breadth and depth to the research inquiry [29]. Review and approval for this study and all procedures were obtained from the Institutional Review Board (IRB) at Drexel University.

### 2.2. Participants

ResearchMatch^®^, a national health volunteer registry that connects potential research participants with researchers, was used to recruit the participants for this study. The participants were eligible for this study if they were fluent in English, at least 18 years of age, resided in the state of Pennsylvania, had at least a high school education, and had access to a computer. The exclusion criteria included medication use for weight loss, adherence to a restrictive diet, and pregnancy, as these factors may change the usual dietary intake. The individuals who were interested in this study completed a screening survey on REDCap^®^. Once the individuals had been determined to be eligible, they reviewed the consent form and provided consent online as a proxy for written informed consent. In the qualitative phase, all the participants were invited to be interviewed, and 13 responded and were interviewed. The participants received USD 30 for completing the survey and dietary assessments and received USD 20 for completing the in-depth interview. 

### 2.3. Data Collection and Measures

The participants completed a survey via Qualtrics^®^ (Qualtrics, Provo, UT, USA), which measured interactions with nature, NR, and physical activity. Three random 24 h dietary recalls were completed by the participants on non-consecutive days via the Automated Self-Administered 24 h Dietary Assessment Tool^®^ to measure dietary quality [30]. A subset of participants also completed in-depth interviews. Data collection occurred between January and September 2022. 

Interacting with nature was assessed using a survey based on the work of Shanahan et al. (2016) [4]. In our survey, the participants were asked to report what close views of nature they had and if there were any pictures of nature in the room that they spent the most time in at home and work. These questions measured indirect interactions with nature. The participants also indicated if they had any plants or flowers in the rooms they spent the most time in at home and work, indicating incidental interactions with nature. Additionally, the participants were asked to self-report the amount of time that they spent in nature during their daily lives over the previous week. This was used to measure the duration of intentionally interacting with nature. 

The NR scale was used to measure the participants connection to nature [18]. The participants indicate their rate of agreement on 21 items using a 5-item Likert scale. The appropriate items are reverse-scored, and the average was taken to calculate the NR total and subscale (experience, perspective and self) scores. The NR total and component scores range from 0 to 5, and a greater score indicates a greater connection to nature. The NR scale has demonstrated acceptable internal validity [18]. 

The International Physical Activity Questionnaire (IPAQ)–short form was utilized to measure the participants’ level of physical activity. The participants indicated the amount of vigorous- and moderate-intensity physical activity and walking they took part in over the previous seven days. Following scoring published by the IPAQ group, the participants were categorized into low, moderate, and high physical activity groups [31]. This questionnaire has demonstrated acceptable internal validity [32]. 

Using the data collected by three, non-consecutive 24 h dietary recalls, dietary quality was assessed using the Healthy Eating Index-2020 (HEI-2020) score, which measures an individual’s adherence to the Dietary Guidelines for Americans 2020–2025 [33]. This index assesses adequate intake of several items: total fruits, whole fruits, total vegetables, greens and beans, whole grains, dairy, total protein foods, seafood and plant proteins, and fatty acids. In addition, the index assesses the intake of food items that should be consumed in moderation: refined grains, sodium, added sugars, and saturated fats. Each component has a standard minimum score of zero and a maximum score of five or ten. The HEI-2020 total score ranges from 0 to 100, and a greater score indicates greater adherence to the Dietary Guidelines for Americans. 

In-depth interviews used a descriptive approach and a phenomenological orientation and were facilitated by a research team member who is experienced in qualitative research [DS] [34]. These interviews were guided by a semi-structured interview guide with probes and transitions as appropriate. The questions explored factors that related to participants’ food choices and interactions with nature. The interview guide was refined during data collection to ensure that rich data were collected. The interviews took place on HIPAA-compliant Zoom™ (Zoom Technologies, Inc., San Jose, CA, USA), lasted about 30 min, were recorded, and converted to a clean verbatim transcript. 

### 2.4. Data Analysis

Quantitative data analysis was conducted using IBM SPSS Statistics, Version 28.0 (Armonk, NY, USA: IBM Corp). Missing data were determined to be missing completely at random and were handled via list-wise deletion, which excluded five cases. The participants’ demographic characteristics, views of nature, and nature in the home, and IPAQ categories were tabulated as frequencies and percentages. Descriptive characteristics, including mean, median, and range, were calculated for dietary quality, NR (total, experience, perspective, and self), and the duration of intentional interactions with nature over the previous week. Within this article, the normally distributed data are presented as mean and standard deviation, and non-normally distributed data are presented as median and interquartile range. The duration of intentional interactions with nature was presented to participants as categorical options (e.g., 1–2 h). As such, the midpoint of each answer was calculated (e.g., 90 min), and minutes were divided by 60 to calculate hours [4]. 

NVIVO 14 (QSR International Pty Ltd. Version 14, 2023, Burlington, MA, USA) qualitative data analysis software was used to aid in qualitative data management and analysis. Qualitative data analysis consisted of thematic analysis using an inductive approach. Two researchers performed the main coding, and a third researcher resolved the discrepancies [D.S., C.S., and B.J.M.]. The coding process consisted of open coding two interview transcripts, and then creating and refining the codebook using an iterative approach. During the analysis of the final interview, no new codes were identified, ensuring that data saturation was reached. After categorizing the codes, themes were developed. The integration of the data involved merging the quantitative and qualitative data to answer the main research question using joint displays. This type of analysis allows for a more nuanced examination of the research findings compared to the understanding that either type of data could provide alone [35].

### 2.5. Enhancing Validity and Credibility

Several steps were taken to enhance validity and credibility in both phases of this research study. In the quantitative phase, internal validity and reliability was established by using validated instruments [36]. Generalizability was established by having inclusion and exclusion criteria and utilizing validated instruments [36]. In the qualitative phase, credibility was established by triangulating the data with quantitative data [37]. Multiple researchers analyzed the qualitative data, providing dependability [37]. Transferability was assured by providing thick description herein [37]. Confirmability was established by recording all the research activities and triangulating the qualitative and quantitative data [37]. 

## 3. Results

The following section describes the results of our pilot convergent mixed methods study. The quantitative results are presented first, followed by the qualitative findings, and lastly the integrated results, which directly address the study objectives. Twenty-five participants completed the survey and dietary assessment, and a sample of thirteen of those participants completed in-depth interviews. The average age of the participants was 38 years (SD = 10.87 years). Most participants were Non-Hispanic White (80%) and female (68%). In addition, 92% of the participants had a college or graduate degree. Forty-four percent of the participants had an annual household income of USD 65,000–77,999. Table 1 displays the participants’ demographics. 

### 3.1. Quantitative Results

The median (IQR, range) amount of time intentionally interacting with nature over the previous week was 3 h (2.63, 15.75). The participants had a median (IQR, range) NR total score and NR Perspective score of 3.43 (1.07, 2.33) and 3.50 (2.08, 3.17), respectively. The average (SD) NR Experience score was 3.47 (0.73) and NR Self score was 3.86 (0.65). The participants had a mean (SD) HEI-2020 score of 54.48 (10.47), which is close to the average HEI-2020 score of 57 for adults living in the United States [38].

Nearly all the participants (92%) indirectly interacted with nature as they had a view of nature from the room they spent the most time in at home or had pictures of nature (40%). The participants incidentally interacted with nature by having house plants (60%) and flowers (52%) indoors. Sixty-eight percent of the participants were categorized as performing high levels of physical activity. The quantitative results are displayed in Table 2. 

### 3.2. Qualitative Results

Three themes relating to spending time with nature, NR, and dietary choices were identified: (1) the influential role of Nature Relatedness, (2) finding harmony in oneself, and (3) connecting to others through nature. The first and second themes begin to explain the connections among interacting with nature, NR, and dietary choices. While the third theme solely focuses on interacting with nature, we feel it is important to highlight the participants’ experiences and voices. The themes, categories, and exemplary quotes are displayed in Appendix A. 

#### 3.2.1. The Influential Role of Nature Relatedness

NR was a pervasive topic throughout the interviews. The participants expressed emotional connections to nature by using words and phrases such as “*love*”, “*enjoy*”, and “*connects us to mother earth*”. Engaging the senses of touch, hearing, sight, and smell were important parts of interacting with nature for our participants. Touching nature was important for P24’s description of interacting with nature: “*If I am on a sidewalk, that clearly makes me feel differently. Cause there can be trees around me but if I’m walking on a sidewalk, it’s very different to me than going on a hike where you’re walking on a dirt path. … What surface I am on makes a difference*”. P11 said “*I love to go to the beach and read, listen to the water. Toes in the sand. It’s kinda my happy place*”. Looking at nature through a window (indirect interaction with nature) was an important way to be immersed in nature for many of our participants. P23 spoke about how he changed where his desk was when he started working from home, “*I was spending a lot of time like looking sideways out the window. I was like, ‘Wait, I could just change my desk’*”. Viewing nature from inside was formative for P09’s feelings about nature: “*When I was growing up, the house that we lived in, the kitchen faced the outside. And so I used to enjoy washing dishes … to see the different types of birds*”. Smell seemed to be important to evoking memories. P10 detailed the nostalgia that she feels when she “*smell[s] cut grass, I think of my dad cutting grass in the summer while I was playing baseball or something with my neighbors*”.

During the interviews, many participants reflected on memories of being in nature and how it has impacted their lives. For P22, her experiences during childhood made her long to be in nature, “*I grew up being able to wander through nature freely. That has always stuck with me. That’s something I yearn for. I don’t want to be a desk worker, but we have to be as adults. If it was up to me, I would spend all my time in nature*”. Likewise, P31 recalled doing a particular hike with her family “*when [she] was seven and eight and nine*”, and further described that her family still hikes that trail once a year, despite it being across the country. P11 described that being at the beach brought upon feelings of wonder, “*Just nature itself is utterly amazing. … How do we survive here? It’s amazing*”. Interacting with nature, creating and reflecting on memories, and experiencing wonder seemed to build the participants’ NR. 

Having a concern for nature and recognizing one’s impact on nature is part of NR. When discussing what factors impact their dietary choices, a number of the participants talked about how they believed they were making sustainable dietary choices, such as buying local foods and food items with less packaging. For P23, the environmental impact of meat was important to his food choices: “*I try to eat things that have less environmental impact. So, I eat meats like chicken and fish. I’m somewhat aware of the greenhouse gas impact of foods*”. Beef was also discussed by P23 and others as being avoided because of its environmental impact. “*I know beef farms are the biggest issue with climate stuff in terms of food. So, we try not to eat a lot of beef specifically*”, said P24. Local foods, wild caught fish, cage-free eggs, seasonal items, and foods from farmers markets were factors in P07’s food choices that lowered her environmental impact. P09 detailed that food packaging impacted what foods she chose to buy, “*[Plastic and Styrofoam are] going to end up in a landfill someplace. They’re not going to break down. So that’s a concern. I would hate to see Styrofoam end up in the ocean because that will harm the inhabitants of the ocean, because it doesn’t break down*”.

#### 3.2.2. Finding Harmony with Oneself

The factors that influenced interacting with nature and for making certain dietary choices were similar and centered around finding harmony with oneself. Being in nature and making certain dietary choices were ways that the participants could engage in healthy behaviors, promoting their own health. For example, interacting with nature allowed time for the participants to disengage from daily life. P05 detailed that being in nature “*gets me away from the hustle bustle*”. The process of disengaging also promoted mental wellbeing. When describing nature, P05 said, “*it’s my peace. It’s my solace. It’s my centering. It’s my rejuvenation*”. Likewise, P07 said, “*I think just the experience of being outdoors, I think it it’s been great for my mental health. It allows me to move beyond the tasks that I have like lined up and the drag of everyday life and to actually connect with my surroundings. I think that that has been important to me*”. Promoting health through outdoor physical activity was another motivator for intentionally interacting with nature. P10 described that a healthy lifestyle was important for her, “*Physical exercise is another aspect of a healthy lifestyle. So that’s another motivator, just cause I want to be active and improve my running overall*”. 

The participants spoke about multiple dietary factors that they made to promote their personal health, including avoiding processed foods, choosing foods with fewer ingredients, avoiding certain ingredients, and choosing whole foods. P05 described that she had kidney disease, so she avoided certain micronutrients, “*I’m very careful now with how much phosphorus is in, how much salt, how much potassium [are in foods]*”. P10 also described dietary components that she avoided, “*I try to avoid having a super high diet of saturated fats, trans fats, a lot of cholesterol, things like that. I try and stay away from more processed foods and rely more on whole foods that I cook*”.

#### 3.2.3. Connecting to Others through Nature

The participants in this study described how intentionally interacting with nature provided them with an avenue to connect with family and friends. Activities such as walks, water activities, and hiking were described as activities that the participants did with others. P11 described that she “*take[s] a walk every evening with [her] neighbor*”. For P29, making her parents happy was her primary motivator for interacting with nature. “*I know [it] make[s] them happy when I spend time with them out there and like they like gardening. So, it’s something that will make them happy if I help them*”. 

Interacting with nature also connected the participants to previous generations, such as their parents and grandparents. P11 discussed how she and her daughter looked for reminders of her father: “*My father passed away recently, and I like to watch for cardinals or symbols that I think is him. So, my daughter and I will sit and watch and wait for the cardinals to come. Even if it’s from the house, if we’re not outside, we will try to enjoy from inside as much as we can*”. Intentionally interacting with nature also connected individuals with their distant ancestors. When describing activities that she does in nature, P31 detailed that a recent visit to Maine helped her appreciate what her ancestors experienced as pioneers: “*[Maine was] a boating, fishing, water culture. Of course, they would not have known how to work a wagon. They weren’t farmers, so it was interesting to just see what a change they made because they left*”.

### 3.3. Integrated Results

Once the quantitative and qualitative data were separately analyzed, the results were compared to identify how the data converged. In Table 3, we highlight two participants whose quantitative results and interviews varied. A comparison of these cases highlighted that NR may promote interactions with nature and the participants’ beliefs that they were making sustainable dietary choices. Participant 07’s HEI-2020 total score, NR total and component scores, and time spent in nature over the previous week were above the sample average. Being surrounded by a natural area was a factor in where they lived and their mental health was positively affected by interacting with nature. The impact that their food potentially has on the environment was important to them and influenced what food items they chose to buy. Participant 30 reported several quantitative measures (HEI-2020, NR Experience, and NR Self scores, and time spent in nature) below the sample average. In their interview, they did not convey an affinity toward nature and did not express how being in nature may have benefited their health. This participant reported that their schedule was the principal influence on their dietary choices. 

During the analysis of the interviews, we identified a cyclical relationship among interacting with nature; creating or reflecting on memories and experiencing wonder; and NR, which may have ultimately impacted dietary choices (Figure 2a). This may be reflected in the quantitative data as the participants who expressed NR and their belief in making sustainable dietary choices had greater seafood and plant protein HEI-2020 component scores (maximum score = 5) (*Mdn* = 5.00, IQR = 1.23, range = 1.64) compared to the other participants interviewed (*Mdn* = 1.64, IQR = 4.01, range = 4.16) (Figure 2b). This meta-inference highlights that NR may influence the importance of and beliefs in making sustainable dietary choices and the practice of consuming more protein sources that have a smaller environmental impact than ruminants (e.g., cows) [39].

The participants conveyed aspects of NR (experience, perspective, and self) during their discussions of interacting with nature and dietary choices (Figure 3). The box plot represents the scores of each subscale and the illustrative quotes provide context as to how dietary choices and interacting with nature pertain to NR. For example, the participants’ familiarity with nature during their childhood remained an important influence for their desire to continue interacting with nature, which pertains to NR Experience. Further, the participants’ perspective of their impact on nature affected their dietary choices (NR Perspective) and discussed that being connected to nature positively impacted themselves (NR Self).

## 4. Discussion

In this pilot convergent mixed methods study, we explored the connections among interacting with nature, NR, and dietary choices. The quantitative and qualitative findings confirmed and expanded upon each other, allowing for greater depth and breadth. The integrated findings indicated that NR promoted interacting with nature and interest in making sustainable dietary choices. Further, the integrated findings highlight that aspects of NR were pervasive in the participants’ discussions of interacting with nature and dietary choices. 

Through data integration, we were able to identify that NR may play a key role in both interacting with nature and the perceived practice of making sustainable dietary choices. Becoming more physically connected to nature through the senses, creating or reflecting on memories in nature, and experiencing wonder in nature built the individuals’ NR (specifically NR Experience and NR Perspective) and further promoted them to interact more with nature. Others have reported that interacting with nature may be associated with NR [2,21,40,41]. Our research builds upon these previous findings, providing more context to how this relationship may be built and strengthened. 

In our study, we found that NR may promote the interest in making sustainable dietary choices. NR and other measurements of connection to nature have been associated with greater environmental stewardship and concern for nature [18,42,43]. We are not the first to report that connection to nature may promote sustainable dietary choices. Weber et al. (2020) reported that individuals who reported greater NR also reported greater sustainable nutrition behaviors, such as the intention to eat sustainably and positive attitudes toward sustainable eating [44]. Likewise, it has been reported that a greater connection to nature measured by the connection to nature scale has been associated with purchasing seasonal, local, organic, and climate friendly foods, which are aspects of sustainable dietary choices [45,46]. The associations between NR and sustainable food consumption have yet to be explored. We report that the individuals who spoke about their beliefs in making sustainable dietary choices, including eating food that they perceived as having a smaller environmental impact, had greater seafood and plant protein intake. The EAT-Lancet Commission, a group of scientists who aim to improve human health and environmental sustainability, has recommended that protein from ruminants (e.g., cows) should be replaced with proteins from plants and seafood, even though seafood can still have an deleterious effect on the environment [39]. While our work begins to explore how NR is related to sustainable dietary choices, seafood and plant protein intake is not a comprehensive nor sufficient measurement of sustainable dietary intake. More appropriate measurements of sustainable dietary consumption, such as the EAT-Lancet Index [47] or Sustainable HEalthy Diet (SHED) Index [48], would provide a more robust understanding of the relationship between NR and sustainable dietary intake. 

The other insights gained in this study were that the motivators to interacting with nature and making certain dietary choices were similar, centering around being healthy, and that our participants found nature as a place they could connect with others. A plethora of research has identified that interacting with nature and eating healthy foods is beneficial for promoting health [1,2,3,4,5,6,7,9,13,15,16,26,27,28]. Our participants were able to identify how interacting with nature and dietary choices affected their health, bridging these behaviors. Multiple researchers have reported that individuals connect with others in nature [49,50,51,52]. Our research adds to these findings as interacting with nature also allows for individuals to connect with previous generations by reminding the individuals of family members, seeking signs of family members through nature, and gaining an appreciation for their ancestors.

### Limitations

We acknowledge several limitations of this research. First, there was a small sample size of participants consistent with a pilot study [53]. Further, the individuals were only sampled from Pennsylvania, USA. These limitations restrict the generalizability of the results. Future studies should be conducted with a larger sample, allowing for statistical tests such as correlation and regression to be conducted. While our exclusion criteria attempted to exclude individuals who may have had changes to their usual dietary intake, we acknowledge that our participants may have additional conditions or circumstances that change their dietary habits (i.e., sicknesses, eating disorders, etc.). Additionally, in the survey, the participants did not have the option to indicate that they did not spend time in nature over the previous week, as such, we were not able to determine if the participant did not spend any time in nature over the previous week or had skipped the question, which excluded five cases. Despite these limitations, this study utilized a convergent mixed methods design which allowed for the quantitative and qualitative data to be triangulated and merged. This allowed each form of data to explain the results of the other.

## 5. Conclusions

Understanding the connections among interacting with nature, NR, and dietary choices may give us insight into how to promote healthy dietary choices through greater interaction with and affinity toward nature. The results from this pilot mixed methods study indicate that there are connections among the variables of interest and that NR may play a pivotal role in how interactions with nature may promote sustainable dietary behaviors. Future studies should examine if NR mediates or moderates the relationship between interacting with nature and sustainable dietary behaviors.

## Figures and Tables

**Figure 1 ijerph-21-00899-f001:**
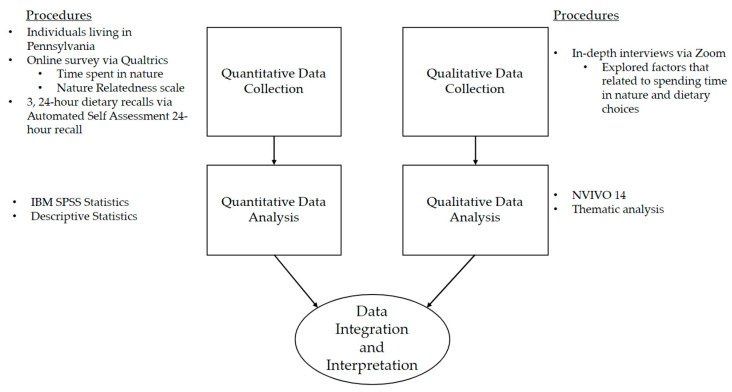
Diagram of convergent mixed methods study design.

**Figure 2 ijerph-21-00899-f002:**
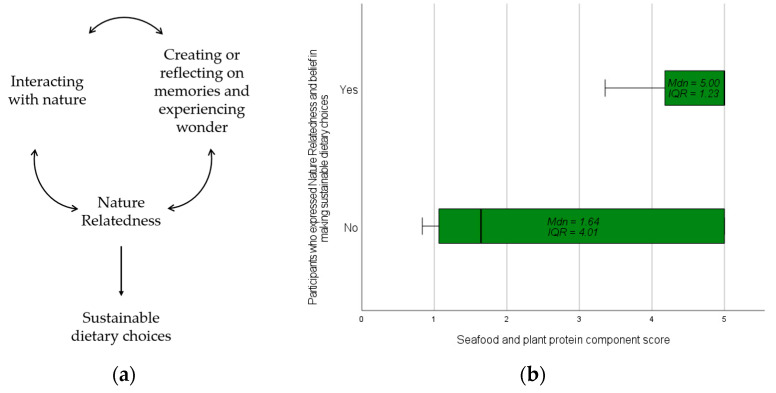
Joint display of the connections among interacting with nature, Nature Relatedness, and beliefs in making sustainable dietary choices: (**a**) qualitative insight; (**b**) reflection of insight in seafood and plant protein component scores.

**Figure 3 ijerph-21-00899-f003:**
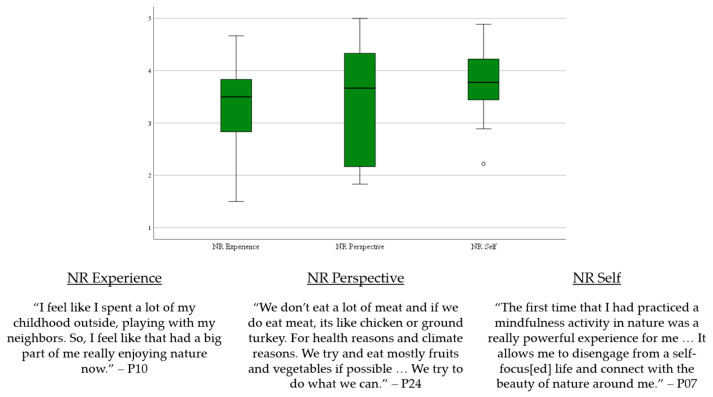
Joint display of Nature Relatedness (NR) subscales and illustrative quotes.

**Table 1 ijerph-21-00899-t001:** Participant demographics ^a^.

	*n* = 25
Age (mean ± standard deviation)	38.00 ± 10.87
Gender	
Male	8 (32%)
Female	17 (68%)
Race and Ethnicity	
Asian	3 (12%)
Black or African American	1 (4%)
Hispanic White	1 (4%)
Non-Hispanic White	20 (80%)
Education	
Highschool, GED, or Some college	2 (8%)
College graduate	14 (56%)
Graduate school	9 (36%)
Annual Household income	
<USD 20,800	3 (12%)
USD 20,800–41,599	1 (4%)
USD 41,600–64,999	4 (16%)
USD 65,000–77,999	11 (44%)
>USD 78,000	6 (24%)

^a^ Values are expressed as number (%) unless otherwise indicated.

**Table 2 ijerph-21-00899-t002:** Key measures.

	M (SD)	Mdn (IQR)	Range
Duration of intentional interactions with nature ^a^	3.94 (3.26)	3.00 (2.63)	15.75
Nature Relatedness Total	3.58 (0.62)	3.43 (1.07)	2.33
Nature Relatedness Experience	3.47 (0.73)	3.50 (1.00)	3.17
Nature Relatedness Perspective	3.26 (1.06)	3.50 (2.08)	3.17
Nature Relatedness Self	3.86 (0.65)	3.78 (0.89)	2.67
Healthy Eating Index-2020	54.48 (10.74)	54.56 (14.25)	43.82
	*n*	%	
Has a view of nature at home (indirect interaction)	23	92	
Types of nature in room spent most time in			
Plants (incidental interaction)	15	60	
Flowers (incidental interaction)	13	52	
Pictures of nature (indirect interaction)	10	40	
Level of physical activity			
Low	2	8	
Moderate	6	24	
High	17	68	

^a^ Represented in hours over the previous week.

**Table 3 ijerph-21-00899-t003:** Joint display of exemplar participants’ quantitative measures, quotes, and mixed methods interpretation.

Participant	Quantitative Results		Exemplar Quotes	Mixed Methods Interpretation
07	HEI-2020	63.40	“I think especially when it comes to things like seafood, like ensuring that it’s like wild caught as opposed to farm raised is important and I think like something that really influences me is trying to buy locally as often as I can. I try to go to farmers markets and buy from local farmers, because I know that even the journey that the food will take up is going to have some type of environmental impact and what farmers end up supplying us tends to be whatever is in season as well, and so I think seasonal foods are also like something that I pay attention to. … I think when it comes to eggs like you know cage free, ensuring that, you know, the chickens are kind of–they have lived like a full life to the extent that I can gather.” “I live in a pretty urban environment … and so I live right across from a park and that was a very important decision to me in like making the decision to live here and in this neighborhood.” “[Nature] has the ability to provide a meditative, self-reflecting quality.”	This participant’s quantitative results were above the sample average. They make specific dietary choices that benefit themselves and believe the dietary choices they make benefit the environment. They value nature, which impacts their daily life and mental wellbeing.
	NR	4.57		
	NR Experience	3.83		
	NR Perspective	5.00		
	NR Self	4.78		
	Duration of intentional interactions with nature	4 h		

30	HEI-2020	46.97	“I see my schedule would be a huge factor in like also availability of food around me. I feel like I wouldn’t go out of my way to probably grab something healthy in that sense.” “I mean, it sometimes reminds me of when I would go to the park as a child with my parents, or even like picnics with my family or this one time I went on a trip with my sister.” “A lot of times I just sit around on the lawn with my friends. I also walk around the river trail. So, I think that is also nice because it is kind of like an enclosed area with a lot of trees and like the rivers just on the side. Yeah, it’s usually just like sitting or like walking around.”	This participant had HEI-2020, NR Experience, and NR Self scores below the sample average. This participant also spent less time in nature over the previous week than the sample. Their personal schedule was the primary factor in their dietary choices; prioritizing easy to grab and quick foods over healthier options. They spoke about childhood nature experiences that they thought about *sometimes* when in nature. While the participant was able to identify nature around them and times that they spend in nature, they did not convey a special affinity toward nature or express how nature benefitted them.
	NR	3.48		
	NR Experience	2.50		
	NR Perspective	4.50		
	NR Self	3.44		
	Duration of intentional interactions with nature	30 min		


## Data Availability

The data that support this article are available from the corresponding author.

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
