# Peer review of "The Connections among Interacting with Nature, Nature Relatedness and Dietary Choices: A Pilot Mixed Methods Study"

_ijerph, 2024, doi:10.3390/ijerph21070899_

Round 1

Reviewer 1 Report

Comments and Suggestions for Authors

It is a very small group of participants and not a balanced group. Only 13 for the in-depth interviews.

There is no discussion og the participants knowledge about dietary recommendations or environmental issues. Therefore, it is very difficult to know if it is the interacting with nature, there knowledge about healthy diet and sustainability or a combination.

The results can only indicate and therefore not be used for the purpose of this study (page 2 line 66-69).

Author Response

Comments and Suggestions for Authors

It is a very small group of participants and not a balanced group. Only 13 for the in-depth interviews.

Thank you for your comment. While this is a small group of participants for the quantitative portion of the study, the sample size is consistent with pilot testing. The sample size allowed us to engage with each participant in a detailed way to explore the connections between interacting with nature, Nature Relatedness, and dietary choices; and inform data interpretation and follow up research studies. We do address the small sample size in the limitations section (page 12, lines 400-401). Having a small group of participants in this study allowed us to test our methods for a larger study. While some convergent mixed methods studies have equal numbers of participants, this is not required. It’s important to highlight that the participants who completed the interview (n=13) also completed the survey (n=25) (Creswell and Plano Clark, 2018). Recommended sample sizes in qualitative work range from 10-20 participants with the goal of reaching data saturation (Guest et al. 2006). We highlight on page 4, lines 164-165, that we achieved data saturation in our interviews. 

Creswell JW, Plano Clark VL. Designing and Conducting Mixed Methods Research. 3rd ed. SAGE publications, Inc.; 2018.

Guest G, Bunce A, Johnson L. How Many Interviews Are Enough? : An Experiment with Data Saturation and Variability. Field Methods. 2006;18(1):25. doi:10.1177/1525822X05279903

There is no discussion og the participants knowledge about dietary recommendations or environmental issues. Therefore, it is very difficult to know if it is the interacting with nature, there knowledge about healthy diet and sustainability or a combination.

The results can only indicate and therefore not be used for the purpose of this study (page 2 line 66-69).

Thank you for your comments. Much research has provided evidence to positive relationships between interacting with nature, Nature Relatedness, mental and physical health and physical activity, but we are just beginning to explore how interacting with nature, Nature Relatedness, and dietary choices are connected. Throughout the manuscript, we are careful to not use language that would suggest quantitative relationships or causation as this was not the purpose of our study, we aimed to use language that suggests how these variables of interest may be connected. Additionally, we did not ask participants about their knowledge of dietary recommendations or environmental issues. In our interview guide, you will see that we asked participants about factors in their food choices. Participants voluntarily provided their beliefs in making sustainable dietary choices.

Reviewer 2 Report

Comments and Suggestions for Authors

This is a sufficiently sound contribution. The research problem is relevant and important. The applied method is acceptable. It would be much better to present a standard-scale quantitative survey than just a pilot study, but the approach is original and valuable enough to justify publication. At least you present a study design that can be replicated in the full scale either by yourselves or some other authors in future.

Please make it more clear in the manuscript what is the role of the qualitative part of your study and what are the main lessons learnt from the interviews for the full-scale research design. I mean what is the differential impact of this part of your pilot study? Your survey design is already quite clear and complete, so what is the use of the in-depth interviews?

Author Response

This is a sufficiently sound contribution. The research problem is relevant and important. The applied method is acceptable. It would be much better to present a standard-scale quantitative survey than just a pilot study, but the approach is original and valuable enough to justify publication. At least you present a study design that can be replicated in the full scale either by yourselves or some other authors in future.

Thank you for your comment. Much research has provided evidence to positive relationships between interacting with nature, Nature Relatedness, mental and physical health and physical activity, but we are just beginning to explore how interacting with nature, Nature Relatedness, and dietary choices are connected. As both quantitative and qualitative researchers, we understand the strengths of both methods and wanted to use both to complement each other, providing more breadth and depth to this novel research inquiry (page 2, lines 76-78). Completing this pilot study also allowed us to test and refine our methods for a larger mixed methods study as the results presented herein point us in future directions.  

Please make it more clear in the manuscript what is the role of the qualitative part of your study and what are the main lessons learnt from the interviews for the full-scale research design. I mean what is the differential impact of this part of your pilot study? Your survey design is already quite clear and complete, so what is the use of the in-depth interviews?

Thank you for your questions. As mentioned above, we utilized a mixed methods design to utilize the strengths of quantitative and qualitative methods. Interviewing our participants allowed us to learn more about participants personal experiences related to our research question and provided more context to the quantitative data. You can see that we utilized the participants voices throughout the integrated results section (pages 8-11). In Table 3, we compare both the quantitative and qualitative results between a participant who reported measures above the sample average, and a participant who reported measures below the sample average. The quotes from their interviews provided greater context to their lived experiences with nature, Nature Relatedness, and dietary intake. Figure 2 highlights that during analysis of the interviews, we identified a cyclical relationship between interacting with nature; creating and reflecting on memories; and Nature Relatedness, which may have affected dietary choices. This led us to turn to our quantitative data to compare the seafood and plant protein intake of interviewees who expressed greater Nature Relatedness and beliefs in making sustainable dietary choices. Without the interviews, we would not have been prompted to look at the dietary intake of those participants in that way. Finally, Figure 3 displays exemplary quotes that fit into the three subscales of Nature Relatedness. Nature Relatedness is a novel concept and using the quotes allows us as learners to see how the different parts of Nature Relatedness relate to interacting with nature and dietary intake. Throughout these three displays of data integration, we see the belief in making sustainable dietary choices (which was learned from the participants voices) and we will be objectively measuring sustainable dietary patterns in a future study, which we highlight the need for on page 12, lines 384-387.

Reviewer 3 Report

Comments and Suggestions for Authors

It is a novel work and provides information of interest in the area of ​​food and interaction with the environment.

Comments:

1.       How did the authors eliminate or reduce lying bias in the R24 hrs self-administered questionnaires? “Three 24-hour dietary re-96 calls were completed by the participants on non-consecutive days via Automated Self-Administered 24-hour Dietary Assessment Tool® to measure dietary quality”

2.       How many participants entered the study initially? the same 25 subjects who completed the study?

3.       What criteria did you use to interview the 13 subjects in the in-depth interview? comment and describe

4.       Is it possible that one of your participants had an eating disorder? Was there any exclusion criterion that would have identified this type of disorder? If not, it should be identified as a weakness of the study

Author Response

It is a novel work and provides information of interest in the area of food and interaction with the environment.

Comments:

  1. How did the authors eliminate or reduce lying bias in the R24 hrs self-administered questionnaires? “Three 24-hour dietary re-96 calls were completed by the participants on non-consecutive days via Automated Self-Administered 24-hour Dietary Assessment Tool® to measure dietary quality”

Thank you for your question. With any self-reported dietary assessment, we are aware of and try to minimize bias in several ways. First, ASA24 is a tool that utilizes the multiple pass method where the participants first report the foods that they consumed, then provide details to those items, and finally are prompted with commonly forgotten food items. ASA24 provides appropriate probes to the participant, which allows them to enter additional foods consumed multiple times during the recall. Second, when conducting 24-hour dietary recalls in person, participants may misreport what they consumed to “please” the researchers. Using ASA24 potentially reduced this bias as participants are reporting their intake directly into a digital device and are not interacting with research staff during the recall. Finally, these recalls were administered to the participants randomly, meaning that the participants did not know when they were going to be asked to complete a recall. We have added this detail to page 3, line 101. 

  1. How many participants entered the study initially? the same 25 subjects who completed the study?

Thank you for your question. 30 participants entered this study. We were unable to analyze data from 5 of the participants because they had missing data which was missing completely at random (page 4, lines 148-149). We have also edited Figure 1 by removing the sample sizes in each phase (page 3).

  1. What criteria did you use to interview the 13 subjects in the in-depth interview? comment and describe

Thank you for your question. After reviewing our notes, all the participants were invited to participate in the interview portion of the study but only 13 responded and conducted interviews. We have clarified this on page 3, lines 94-95.

  1. Is it possible that one of your participants had an eating disorder? Was there any exclusion criterion that would have identified this type of disorder? If not, it should be identified as a weakness of the study.

Thank you for this comment. There is certainly a possibility that a participant in our study did have an eating disorder. We did not screen for eating disorders, as this is outside the scope of our study. We have reviewed the data and did not observe any extreme outliers in dietary intake. We have added this to our limitations section (page 12, lines 404 - 407)